# Exploring the Integration of Unmanned Aerial System Technologies into Stormwater Control Inspection Programs

**Jarrell Whitman** [1,*]🆔, **Michael Perez** [2]🆔 and **Roy Sturgill** [3]🆔

1   Department of Biosystems Engineering and Soil Science, University of Tennessee-Knoxville, Knoxville, TN 37996, USA
2   Department of Civil and Environmental Engineering, Auburn University, Auburn, AL 36849, USA; mike.perez@auburn.edu
3   Department of Civil, Construction, and Environmental Engineering, Iowa State University, Ames, IA 50011, USA; sturgill@iastate.edu
*   Correspondence: jwhitma4@utk.edu

**Abstract:** Construction stormwater best management practices and post-construction stormwater control measures are controls and techniques designed to manage and treat stormwater runoff. Departments of Transportation (DOTs) within the United States rely on these practices to treat and improve water quality emanating from DOT rights of way. To ensure operational performance, these practices undergo periodical inspections to identify if operational deficiencies exist and if corrective measures need to be deployed. The inspection process is often conducted on foot by a qualified inspector and can require a substantial labor effort to complete. Recently, unmanned aerial system (UAS) technologies have been utilized in the construction sector to survey, monitor, and improve safety. This study sought to identify and document practices regarding UAS technologies when conducting inspections of stormwater practices. Through a distributed DOT survey questionnaire (80% response rate) and four case example interviews, this study investigates how UAS stormwater inspections have been deployed by DOTs and the strategies and programs that have been adopted or created. Key findings outline (1) use of UAS technologies for stormwater inspections, (2) applying UAS technologies within a DOT, (3) staffing and equipping needs, and (4) managing UAS inspection datasets. The study also identifies challenges and implementational strategies to facilitate the development of a UAS stormwater inspection program within a DOT.

**Keywords:** construction; post-construction; stormwater; inspections; monitoring; technologies

## 1. Introduction

Departments of Transportation (DOTs) are tasked with constructing, inspecting, maintaining, and operating a vast array of transportation networks that citizens rely on daily. In doing so, DOTs must consider environmental impacts associated with each of these tasks, primarily those associated with stormwater runoff. The Clean Water Act of 1972 was enacted to establish water quality standards throughout the United States and limit pollutant discharges into the nation's water resources [1]. Major contributors impacting the quality and quantity of stormwater effluent can be linked to infrastructure development which has been fueled by population growth. DOTs rely on best management practices (BMPs) to manage stormwater runoff during construction activities. BMPs are primarily implemented to minimize sediment discharge from disturbed areas. Post-construction stormwater control measures (SCMs) are implemented by DOTs to manage stormwater runoff from impervious areas. These practices primarily target water quality and quantity and function by settling suspended materials, infiltrating runoff, filtering pollutants, or promoting evapotranspiration and water reuse. Many research studies have shown that stormwater BMPs and SCMs can be highly effective means for mitigating undesirable impacts associated with stormwater effluent [2–8]. Major benefits outlined by the United

States Environmental Protection Agency (USEPA) of effective BMPs include protection of wetland and aquatic ecosystems, improved quality of receiving waterbodies, conservation of water resources, protection of public health, and flood control [9].

With increasing environmental regulatory oversight, many DOTs have sought to improve their standardized BMP and SCM design standards and specifications by scientifically evaluating their performance and testing innovative design modifications. In doing so, DOTs have also developed comprehensive inspection and maintenance protocols to facilitate longevity and resiliency of BMP and SCM assets. As part of these efforts, several DOTs have investigated the application of unmanned aerial systems (UASs) when monitoring and inspecting stormwater BMPs and SCMs throughout their life cycle [10,11].

## 2. Objective

Inspection of stormwater BMPs and SCMs can be quite labor intensive to implement, and strategies can vary widely among DOTs [12]. Thus, the primary objective of this study was to investigate the application of UAS technologies by DOTs when conducting inspections of stormwater BMPs used during construction and SCMs used to manage post-construction stormwater runoff. Additionally, this research effort sought to identify innovative UAS stormwater inspection practices implemented by DOTs, UAS data storage and processing strategies, benefits and unexpected challenges encountered, and how DOTs foresee these technologies being utilized for stormwater inspections in the future. Key findings from a distributed survey are presented along with an analysis of four case example interviews.

## 3. Background

Stormwater BMPs and SCMs are practices designed to minimize the environmental impacts of stormwater runoff by capturing, treating, and/or infiltrating stormwater before it enters natural waterways. Stormwater BMPs are crucial in reducing the amount of pollution and erosion that occurs during heavy rain events. However, the effectiveness of BMPs/SCMs can vary depending on the type of practice, site conditions, service level, maintenance, and other factors. A literature review of stormwater practices reveals a wide range of measures and technologies that can be implemented to effectively manage stormwater.

### 3.1. Construction Stormwater Management

Throughout the construction process, erosion and sediment control practices (e.g., BMPs) are used to minimize soil loss from a site and capture suspended sediment prior to offsite discharge. For these BMPs to be effective throughout the project life cycle, contractors must properly install, inspect, and provide maintenance. Some of the most common types of structural sediment control BMPs utilized within the construction industry include sediment barriers, check dams, inlet protection practices, and sediment basins. While these temporary BMPs each have unique design features to optimize performance, their overall objective is to remove suspended soil particles from stormwater runoff through the process of sedimentation and filtration [13,14].

By far, the most effective BMP to control erosion is the preservation of existing vegetation on a site. Because this is not always possible due to site topographical designs and earth-moving requirements, the re-establishment of vegetation becomes a critical step in the construction process. This type of non-structural BMP can consist of temporary and permanent seeding, sodding, straw mulching, and/or rolled erosion control products. In combination, these components conserve soil from splash erosion caused by raindrop impacts. To promote establishment, contractors must select the proper vegetative species for the region and season, employ effective sowing methodologies, add soil amendments as prescribed by a proper soil analysis, provide adequate cover, and maintain adequate soil moisture.

Implementation of construction BMPs is mandated by the USEPA's Construction General Permit (CGP). Construction sites disturbing one or more acres (0.4 or more hectares)

must implement a comprehensive stormwater pollution prevention plan that includes the use of BMPs to minimize erosion, capture sediment, and minimize the transport of other construction related pollutants. The CGP requires routine and regular inspections of BMPs to ensure proper implementation, performance, and to provide corrective action as needed [9].

### 3.2. Post-Construction Stormwater Management

Once construction activities are complete, post-construction SCMs, also referred to as green infrastructure (GI) or low-impact development (LID) practices, are used to mitigate environmental impacts associated with urbanization. Common environmental concerns associated with urbanization include increased stormwater runoff volumes, higher peak flow rates, and physical, chemical, and biological pollutants. Many post-construction SCMs developed utilize natural processes such as evapotranspiration, infiltration, filtration, and/or water reuse to restore natural hydrology and capture contaminants. Examples of post-construction SCMs include bioretention cells, infiltration swales, detention and retention basins, constructed wetlands, filter strips, as well as proprietary practices [15,16].

Post-construction SCMs are mandated through Municipal Separate Storm Sewer System (MS4) permits. These permits require entities classified as MS4s to implement SCMs that protect receiving water bodies from runoff quality and quantity. Almost all State DOTs are regulated through MS4 permits (or similar Transportation Separate Storm Sewer System (TS4) permits) and must include the use of permanent SCMs when adding additional impervious areas. MS4 permits further require the routine inspection and maintenance of SCMs to ensure performance.

Despite the benefits of post-construction SCMs, there are several barriers to their implementation, including lack of funding, space availability, and regulatory barriers. Efforts to address these barriers include providing incentives for SCM implementation and encouraging the development of innovative practices that can be implemented in smaller spaces. The inspection requirements for these measures vary depending on local regulations, the specific type of SCM, and the site's characteristics. Additionally, many DOTs struggle to meet inspection and maintenance requirements for existing post-construction SCMs. This can be attributed to a lack of DOT stormwater staff to conduct inspections, as well as a shortage of DOT maintenance staff capable of conducting routine and non-routine maintenance activities.

### 3.3. UAS Stormwater Inspections

Stormwater BMP and SCM inspection and maintenance activities are crucial aspects associated with performance, as poorly maintained practices can become ineffective and even cause damage to the environment. Inspections are typically conducted by a qualified inspector, and they are usually required by local, state, or federal stormwater regulations. Prior to conducting an inspection, the inspector will often review stormwater design drawings/specifications and develop an inspection plan. Once on site, inspectors will confirm BMPs detailed within design drawings are in the proper location, installed according to specifications, and functioning as intended. Deficiencies observed by the inspector are noted in an inspection report and contractors are notified of corrective actions required.

Traditionally, construction BMP inspections are conducted on foot which can become quite time consuming on large, linear projects. UASs have quickly emerged as a tool to assist DOTs across a variety of inspection and operation applications. Perez et al. [17] conducted a study that explored the application of UAS technologies for conducting erosion and sediment control site inspections during the construction process. Results suggested that UAS technologies can be effective tools for data acquisition during an inspection, but additional software applications would be needed to efficiently incorporate UAS datasets into inspection reports accepted by regulatory agencies. To bridge this knowledge gap, Kazaz et al. [18] conducted a study that utilized UAS dataset imagery to construct a model that was then analyzed by a deep-learning-based object detection system. The detection

system was able to accurately identify four different types of construction BMPs with 100% accuracy on the mean average precision. In addition to these research efforts, several additional studies have focused on current performance, improving performance, and applications of UAS technologies with DOT applications. Table 1 provides a summary of UAS DOT studies conducted over the past decade, as well as the focus area of the study.

**Table 1.** Summary of UAS DOT studies conducted over the past decade.

| Study | Area of Focus |
| --- | --- |
| McDonald [11] | Urban Stormwater Management |
| Perez et al. [17] | BMP Inspections on Construction Sites (TRR) |
| Kazaz et al. [18] | BMPs on Construction Sites |
| Snyder et al. [19] | Summary of State DOT Use (NCHRP) |
| Rogers [20] | Construction Inspections (FHWA) |
| Harper et al. [21] | Technologies for Construction Delivery (NCHRP) |
| Gheisari and Esmaeili [22] | Construction Safety |
| Alexander et al. [23] | Culvert Inspections |
| Turkan et al. [24] | Summary of Use on Highway Construction (NCHRP) |
| Whitman et al. [25] | UAS for Stormwater BMP Inspections (NCHRP) |

## 4. Methodology

The objective of this study was to examine DOT applications of UAS technologies as a tool for inspection of stormwater BMPs and SCMs. To achieve the objective, the researchers employed a four-phase research methodology. Each phase is further explained below.

### 4.1. Phase 1: Survey Development

Phase one of the research methodology focused on using findings from reviewed literature to develop a web-based questionnaire survey. Literature was obtained from a wide range of databases and search tools including, but not limited to: Transportation Research Information Documentation (TRID), American Society of Civil Engineers (ASCE) libraries, Federal Highway Administration research library, and Google Scholar. The questionnaire contained 32 possible questions that were subcategorized into the following six areas of interests:

- Use of UAS for environmental assessments, permitting, and/or stormwater inspections;
- Planned development of a UAS BMP/SCM field inspection program;
- State of the practice for DOT UAS use in BMP/SCM field inspections;
- State of the practice regarding staffing, equipment, and training;
- State of the practice regarding software, data use, and processing; and
- Challenges and benefits.

The survey began with a cover letter that described the purpose of the questionnaire, tips and guidance for completing the survey, as well as key definitions to align terminology. The survey was designed with question skip logics to optimize data collection based on answers provided by recipients on current practices within their DOT.

### 4.2. Phase 2: Recipient Identification and Survey Distribution

To develop a targeted distribution list of DOT survey recipients with stormwater and/or UAS backgrounds, the authors reviewed rosters of the Transportation Research Board's (TRB) AKD50 Standing Committee on Hydrology, Hydraulics, and Stormwater, the American Association of State Highway and Transportation Officials (AASHTO) Technical Committee on Hydrology and Hydraulics, as well as the attendees of the 2022 National Stormwater Practitioners Forum, and by searching individual DOT stormwater management web pages for appropriate DOT staff members. This led to the development of a distribution list that included one recipient from each DOT within the United States, as well as the District of Columbia. The survey was launched on 27 April 2022 and responses were gathered through June 2022, with the majority of responses collected by 13 May 2022.

After sending email reminders and contacting unresponsive recipients via phone call, a total of 41 responses were received, representing an 80% response rate. Figure 1 illustrates states that responded to the survey.

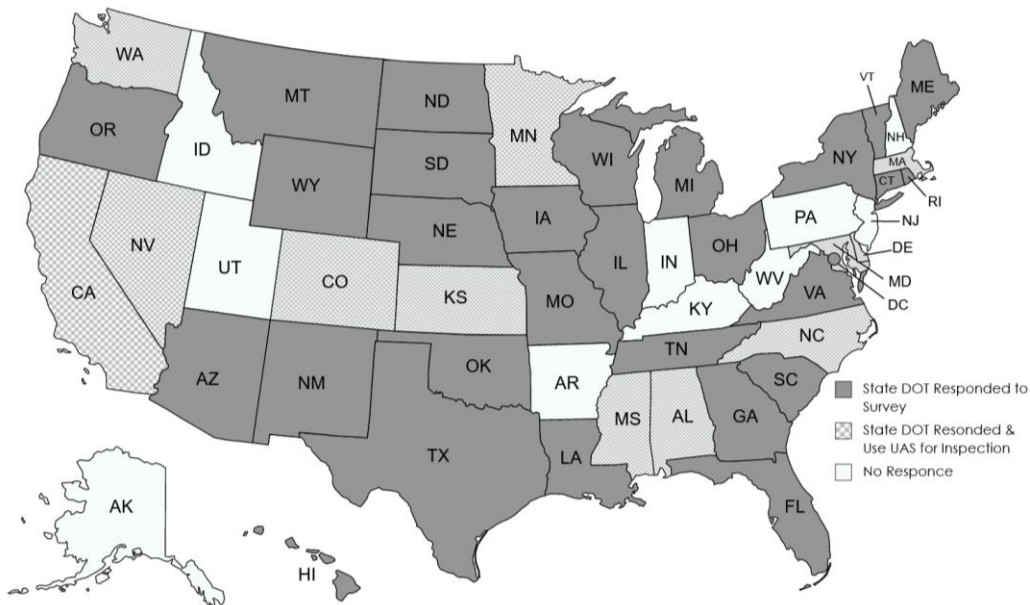

**Figure 1.** Map of State DOTs responding to survey.

### 4.3. Phase 3: Data Sorting and Analysis

Survey data associated with each survey question were sorted at an aggregated level to facilitate understanding, analysis, and visualization of the results. The primary methods for data sorting and visualization were bar percentage plots and pie charts. Plots and charts were then analyzed to identify patterns and trends within the data. The culmination of these patterns and trends represents the state of the practice for UAS as a tool for stormwater BMP and SCM inspections within DOTs.

### 4.4. Phase 4: Case Example Interviews

Survey data collected through the questionnaire were used to draw conclusions on the implementation and challenges associated with using UAS technologies for conducting stormwater inspections. Additionally, the data were used to identify DOTs that utilize UAS technologies for stormwater inspections more frequently than others and that have experience managing UAS inspection datasets. In total, seven US State DOTs were invited to participate in case example interviews; however, only four DOTs accepted the invitation. The researchers conducted in-depth case example interviews to gather specific information on the use of UAS technologies during stormwater inspections. The four US State DOTs that agreed to participate in the case example interviews were: Alabama, Colorado, Delaware, and Kansas.

### 5. Findings

Assessing the state of the practice within DOTs can be extremely beneficial as it provides a means for reviewing and disseminating the latest trends, techniques, and best practices to peer agencies. Additionally, it can assist DOTs in identifying areas for improvement and strategies to reduce expenditures while maintaining an effective level of service to infrastructure needs. The following is a summary of key finds obtained from the literature review, survey questionnaire, and case example interviews.

### 5.1. Use of Technologies for Stormwater BMP and SCM Inspections

Nineteen (46%) of the forty-one DOTs that responded to the survey currently utilize UAS technologies to conduct environmental site assessments and/or permitting on DOT projects; of which, twelve (29%) are specifically using them to conduct stormwater inspections in some capacity. Upon further investigation, these 12 (29%) agencies had chosen to implement the technology voluntarily due to interest from an internal champion. Of the DOTs not currently using UAS technologies for stormwater inspections, 20 (49%) indicated that they were interested in developing a UAS stormwater inspection program, 7 (17%) were not interested, and 2 (5%) were in the developmental phase. Figure 2 summarizes the status of UAS-based stormwater inspection programs among the surveyed DOTs. Of particular interest were DOTs interested in developing a UAS stormwater inspection program. Further analysis of these DOTs suggested that there is a lack of guidance and understanding on how to effectively implement and utilize UAS technologies and associated software platforms when conducting a UAS-based stormwater inspection, especially if an in-house UAS division is not readily available.

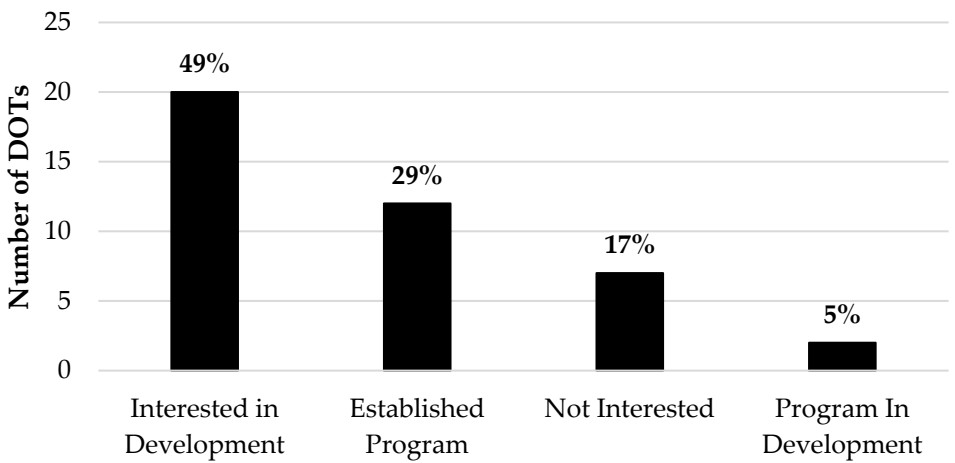

**Figure 2.** DOT UAS stormwater inspection program status.

### 5.2. Applying UAS Technologies for Stormwater Inspections

The next segment of survey questions was presented to DOT respondents who indicated their DOT had an established UAS stormwater inspection program. Of particular interest was the frequency in which a DOT utilizes UAS technologies to conduct a stormwater BMP or SCM inspection. As shown in Figure 3, nine (75%) of the twelve DOTs rarely utilize UAS technologies with only one DOT (Alabama) utilizing the technology 100% of the time. Further investigation into these findings suggested that the primary factors hindering utilization of UAS technology were lack of trained/qualified personnel within the DOT to conduct the UAS inspection and specific environmental regulatory requirements mandating traditional on-foot inspections be conducted on stormwater BMPs/SCMs. Respondents noted that specific factors that often trigger the use of UAS technology, typically in combination with on-foot inspections, include sites with limited/restricted access and environmentally sensitive sites.

The survey also aimed to determine at what points during or after the construction process UAS technologies are most likely to be deployed. The results showed that 83% of DOTs primarily use UAS technologies soon after stormwater BMP installation on a project site, mainly to verify the installation of the practices. Other inspection time intervals (e.g., weekly, bi-weekly, monthly, annually, etc.) were available for selection in the survey but were not notably selected by DOT respondents. The most commonly inspected BMPs during construction were sediment basins/traps (58%) and erosion control blankets/turf reinforcement mats (58%), while the most commonly inspected post-construction SCMs were detention basins (58%) and infiltration basins/trenches (58%). As illustrated in

Figure 4, specific performance elements commonly assessed using UAS technologies include vegetation establishment (75%), soil erosion (75%), and sediment deposition (75%). These findings suggest that UAS technologies are primarily used to (1) verify BMP installation as shown on the stormwater pollution prevention plan and (2) assess elements easily seen from an aerial perspective.

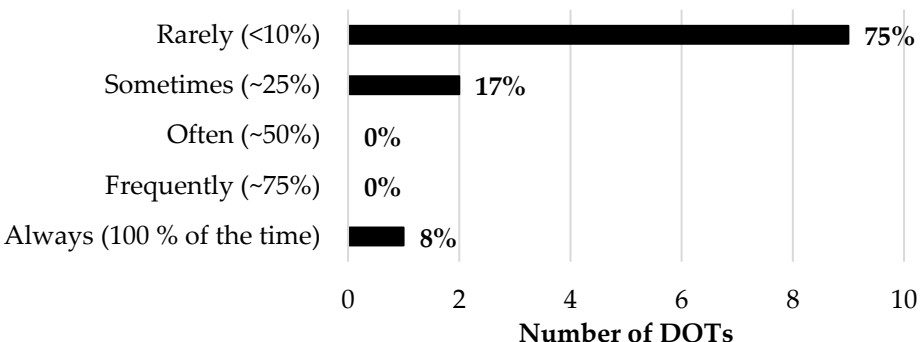

**Figure 3.** Utilization frequency of UAS technologies for BMP and SCM inspections.

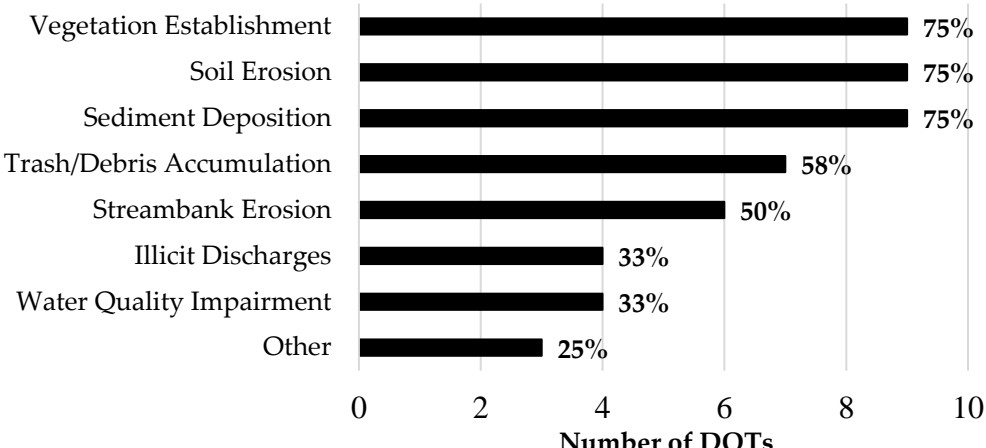

**Figure 4.** Stormwater inspection elements assessed using UAS technologies.

*5.3. Staffing and Equipping a UAS Stormwater Inspection Program*

The survey also included questions regarding the staffing and equipment needs for operating a UAS stormwater BMP inspection program within a DOT. The results showed that 8% of the respondents rely solely on consultants/contractors to conduct UAS stormwater inspections, 34% rely solely on in-house personnel (i.e., DOT staff), and the remaining 58% utilize both consultants/contractors and in-house personnel. Regarding UAS stormwater BMP and SCM inspection training or instructional resources, 37% of DOT staff respondents reported having resources available, 36% had no resources available but expressed a desire to develop them, and 27% had no educational content at all. These results suggest that there is a need among DOTs for the development of educational materials and outreach training initiatives to optimize the applications of UAS technologies during stormwater BMP and SCM inspections. Such initiatives could help address knowledge gaps or skills shortages among in-house personnel and contractors/consultants and ensure that UAS technologies are being used safely and effectively.

The application of UAS platforms and sensors in construction has indeed become increasingly popular over the past decade, particularly for surveying, inspecting, and monitoring purposes. Because of this, the survey sought to determine which UAS systems or platforms are commonly used for stormwater BMP inspections, as well as preferred sensor technologies for data acquisition. The survey found the majority (73%) of DOTs solely utilize rotary airframes, such as quadcopters, for stormwater BMP inspections. The

remaining 27% deploy both rotary- and fixed-wing airframes. Additionally, the optical camera was reported as the primary sensor type used for data collection, with 100% of respondents indicating its use. Other sensors that were noted to be deployed for stormwater BMP and SCM inspections include LiDAR (27%), thermal/infrared (18%), and multispectral (18%). When acquiring UAS equipment for stormwater BMP and SCM inspections, the primary factors to consider are cost and regulations (i.e., FAA, State, etc.), according to 91% of respondents. This highlights the importance of not only considering the capabilities and features of the UAS platform and sensor technologies but also the financial and legal implications of their use. Overall, the results of the survey highlight the importance of not only investing in the appropriate UAS technology for stormwater BMP and BMP inspections but also ensuring that personnel have the necessary knowledge and skills to use this technology effectively.

*5.4. Managing UAS Stormwater Inspection Datasets*

Managing UAS datasets can be a complex task that involves several stages, including processing, analysis, and storage. To develop a better understanding of these processes, the survey sought to identify how DOTs use UAS stormwater inspection datasets. Results indicated that 75% of DOTs primarily use collected data, usually in the form of aerial imagery, to monitor stormwater BMP and SCM maintenance needs. By incorporating the aerial imagery into inspection reports, inspectors can communicate their findings more clearly to maintenance personnel and contractors, which can help ensure that the necessary maintenance work is carried out promptly and effectively.

While digital terrain and surface models can be valuable tools for the construction industry, they are not widely used in the context of stormwater inspections, as noted by 78% of respondents. One possible reason for this could be that the information provided by these models is not considered essential for monitoring and maintaining stormwater BMPs and SCMs. It is also possible that the development and use of these models require specialized skills and software that may not be readily available to DOT stormwater inspection teams. It is worth noting that digital terrain and surface models can provide valuable information on topography, elevation, and surface features, which can help in the design and construction of stormwater practices. By analyzing these models, engineers and designers can identify areas that are prone to erosion or flooding, which can help them design stormwater BMPs and SCMs that are more effective in managing stormwater runoff. Additionally, these models can be used to simulate the effects of various storm events on the landscape, which can help engineers and designers develop more accurate and effective stormwater management plans.

Data storing and management was ranked (3.3 out of 5.0) as the most difficult challenge associated with UAS stormwater inspections. Datasets need to be properly organized and accessible to the appropriate stakeholders, which can be a challenge without the right systems and processes in place. Nonetheless, 82% of the respondents indicated that their DOT had developed a means for incorporating inspection datasets into an existing asset management system/platform. By integrating data from aerial inspections into an existing asset management system, DOTs can more easily track the condition of their stormwater BMPs and SCMs and prioritize maintenance activities. This can help allocate resources more effectively and ensure that practices are functioning as intended.

Artificial intelligence (AI) refers to the development of computer systems that can perform tasks that typically require human intelligence, such as learning, reasoning, and decision making. AI algorithms are designed to analyze large amounts of data and identify patterns and relationships, enabling them to make predictions and decisions based on that data. Respondents were asked if their DOT is using, or planning to use, AI algorithms to estimate quantity, overall condition, status, and/or locations of stormwater BMPs and SCMs installed on site. Results indicated that 45% of DOTs were using, or planning to use, AI technology. By using AI algorithms, DOTs can more easily track the condition of their BMPs, identify maintenance needs, and prioritize repairs and upgrades. Additionally, AI

can help DOTs predict the performance of their stormwater assets under different weather conditions or land use scenarios, enabling them to better plan and design their stormwater management systems.

*5.5. DOT Case Example Interviews*

Data from the questionnaire were used to select DOTs for case example interviews. DOTs were qualified for case example interviews based on their experience using UAS technologies for stormwater BMP and SCM inspections, data application and implementation strategies, comprehensiveness, and willingness to participate in the interview process. Open-ended follow-up interview questions were attached to an email invitation. Follow-up interview questions were used to guide the discussion and provide an avenue to explore unique aspects portrayed by each DOT during the interview. The four U.S. State DOTs that agreed to participate in the case example interview were: Alabama, Colorado, Delaware, and Kansas.

### 5.5.1. Alabama DOT

The Alabama DOT is using UAS technologies to conduct construction stormwater BMP inspections. They have a dedicated UAS Section housed within the Maintenance Bureau, which investigates applications of UAS technologies for transportation engineering, construction, aerial mapping, and utilities. The Environmental Construction Section within the Construction Bureau mandates the use of UAS-acquired imagery for documentation pre- and post-construction and for periodic reviews of construction activity throughout the course of a project. The Construction Bureau has become the largest end-user for the UAS Section that performs all flights for the Alabama DOT. In general, the UAS inspection process involves pre- and post-construction inspections using a fixed-wing UAV to gather video datasets of the construction site to record existing conditions, including BMPs used during construction along with any associated deficiencies. UAS-based inspections are conducted at least once every six weeks using a rotary-style UAV. Collected imagery is used to supplement inspection findings from regulatory mandated on-foot inspections of BMPs. The UAS Section collects 360-degree panorama imagery, which is then uploaded and stored for at a period of at least three years in a cloud-based photogrammetry and mapping software platform that can be accessed by DOT staff. The personnel from the Alabama DOT Environmental Construction Section reviews the datasets to look for any deficiencies and to document compliance or issues identified on a Flight Review List within the Construction Bureau. The Flight Review List is sent to the assigned Area Stormwater Coordinator so corrective actions can be taken. If needed, a focused UAS inspection can be requested to further inspect critical construction stormwater BMPs.

The Alabama DOT plans to assign a UAS pilot to each Regional Office to increase the frequency of UAS inspections for ongoing projects throughout the state, potentially on a weekly basis. The Alabama DOT is also funding a research and development project that will enable stormwater inspectors to use UAS technologies to identify plant species, density, and health. The goal is to use these UAS inspection innovations to identify if on-site vegetation meets the 85% density requirements set by the Alabama Department of Environmental Management (ADEM) for the termination of permit coverage.

### 5.5.2. Colorado DOT

The Colorado DOT uses UAS technologies for less than 10% of their inspections of stormwater practices. They primarily rely on rotary-wing UAS systems with an RGB camera but are also exploring thermal and multispectral sensors. The inspections are carried out by both DOT personnel and by third-party contractors. The Colorado DOT indicated that UAS-based inspections for stormwater applications can be resource intensive and time consuming. To comply with new guidelines requiring visual assessments of all stormwater BMPs, pollutant sources, and discharge points at least once every 45 days on active sites, the Colorado DOT partnered with a private contractor to explore the development of an AI

application that analyzes UAS-acquired photographs to identify inadequate, deficient, and damaged stormwater BMPs.

The first phase of development focuses on inspecting and identifying damaged sediment fences. Although there is no written procedure yet, the inspection process for the AI system development involves capturing imagery over the same segments of silt fence after qualifying storm events, following the same flight path. Each UAS inspection takes about 15–20 min, during which photographs are collected on the upstream and downstream face of the sediment fence. The camera is angled 45 degrees to gather images. Each photograph includes time stamps and geolocation information. After aerial imagery is captured, the data are transferred from the memory device on the UAS to a laptop computer. The data are then processed on site using the subcontractor's machine-learning system and programming logic. The AI software searches the imagery dataset for silt fence deficiencies, including: missing or damaged fence posts, tears or rips in the silt fence fabric, and excessive accumulation of sediment upstream of the fence. When deficiencies are detected by the software, it extracts geotags from the dataset and uploads location points to a site map showing the where the deficiency is located. Ultimately, the goal is to minimize the duration of traditional on-foot stormwater BMP inspections using the technology.

### 5.5.3. Delaware DOT

The Delaware DOT reported that they use UAS technology in less than approximately 10% of stormwater BMP inspections. The Delaware DOT uses an application software for their stormwater BMP inspections that is capable of being downloaded to a user's smart device when conducting traditional on-foot inspections. The application requires the user to input stormwater BMPs and their locations on the site before conducting an inspection. Once on site, the system directs the user to the BMP locations via the smart device's GPS connection and displays an inspection dashboard based on the type of BMP being evaluated. Once the inspection is complete, the application generates and shares a PDF report, including images, with all parties involved with the project.

Although the DOT prefers using the application software for BMP inspections, the software does not currently incorporate UAS imagery into the generated inspection report. Thus, traditional pen-and-paper inspections are often conducted in combination with UAS inspections for large, linear highway modification and expansion projects. Aerial images are the predominant dataset collected during an inspection flight, with the occasional aerial video. Once the user has completed the UAS-based inspection, the inspector downloads the dataset to a computer. Currently, the Delaware DOT does not have a system to manage data for UAS datasets. Inspectors hand-select aerial images from the dataset and insert them into an inspection report created in a word-processing software. Due to a lack of trained personnel in the use of UAS technologies, the Delaware DOT primarily outsources UAS-based inspections to local consulting firms that have capabilities to conduct stormwater BMP inspections.

### 5.5.4. Kansas DOT

The Kansas DOT uses UAS technologies for stormwater BMP inspections during the construction process less than 10% of the time, primarily using rotary-wing UAS technologies with an RGB sensor. These inspections are performed by DOT personnel, mainly on projects over 100 acres due to limited staff trained in UAS operations and stormwater inspection protocols. The primary factor in conducting an inspection using UAS technology is based on time availability to complete the inspection along with traditional on-foot stormwater inspections. Flights are not pre-planned nor conducted over specific stormwater BMPs, and there are currently no written guidelines for conducting stormwater BMP inspections using UAS applications. Personnel that perform UAS inspections of stormwater BMPs follow an undocumented process that has been self-taught or developed through field practice and are constantly modifying and improving the inspection process.

After a flight, DOT staff upload aerial images to a cloud-based photogrammetry software and process them to create an orthomosaic map, staff then upload data to an online map software to generate an overlay that is georeferenced. Features within the online map software allow personnel to focus in on specific areas with visual stormwater BMP deficiencies so they can compare on-foot inspection notes. Screen shots of deficiencies are taken and included in the stormwater BMP inspection reports. This process has shown to be effective for evaluating areas with severe erosion and sedimentation. Table 2 provides key similarities and differences among the case examples.

**Table 2.** Key similarities and differences among case examples.

| DOT | UAS Inspection Frequency | UAS Equipment \| Sensor | UAS Inspections Conducted | UAS Inspections Conducted on |
|---|---|---|---|---|
| Alabama | 100% | Fixed and Rotary Wing \| RGB Camera | In House | During-Construction BMPs |
| Colorado | <10% | Rotary Wing \| RGB Camera | In House and Third Party | During-Construction BMPs |
| Delaware | <10% | Rotary Wing \| RGB Camera and LiDAR | In House and Third Party | Post-Construction SCMs |
| Kansas | <10% | Rotary Wing \| RGB Camera | In House | During-Construction BMPs |

## 6. Discussion

While many DOTs utilize UASs to support inspection operations to some extent, there are still substantial barriers hindering widespread implementation. The successful establishment of a UAS program requires support from DOT administrators and funding mechanisms to cover essential aspects such as hardware, software, staffing, and training. For DOTs that are already using UAS for other inspection purposes, there can be advantages in leveraging existing resources to conduct inspections of stormwater BMP and SCMs. An excellent example of this approach is the Alabama DOT UAS program, which other DOTs might consider emulating. Establishing a centralized UAS program that serves multiple divisions within the DOT, including the construction division to monitor construction progress and the operations and maintenance division to inspect bridges, among other inspection applications, can lead to more efficient and comprehensive use of UAS resources.

### 6.1. Staffing

Several common barriers can hinder the successful implementation of a UAS stormwater inspection program at a DOT, with the foremost obstacle being widespread shortage of staff. Salaries and benefits offered by the private sector often outcompete DOTs, making it challenging to attract and retain top talent. DOTs often experience a high turnover rate due to retirements, internal promotions, short-term positions, and staff members leaving for higher-paying jobs. Additionally, budget constraints and limited resources can worsen the situation. According to the presented study results, the success of UAS stormwater inspection programs often relies on internal champions. These are stormwater professionals who are enthusiastic about UAS use and can convince management of its numerous benefits. This grassroots effort can be highly effective when a champion exists. DOTs that have implemented UASs for stormwater inspections have experienced notable benefits, and those planning to do the same should consider dedicated funding and positions. However, finding staff with expertise in both stormwater inspections and UAS operation can be challenging due to the specific set of skills and certifications required.

DOTs can employ several strategies to overcome personnel needs for UAS-based stormwater inspections. These strategies may include providing monetary incentives for certification and cross-training in stormwater inspections and UAS operation. UAS programs can also be utilized to enhance recruiting efforts, exciting prospective employees, particularly at the high school and undergraduate levels, and fostering interest in UAS operation and stormwater management careers. Collaborating with local universities, community colleges, and technical programs to offer UAS operation and inspection courses or workshops can be beneficial. Another approach is to offer stormwater training to DOT staff already skilled in UAS applications for other types of inspections. Additionally, DOTs

might consider outsourcing UAS stormwater inspections to local contractors with expertise in the field. By adopting these strategies, DOTs can address personnel challenges and establish effective UAS-based stormwater inspection programs.

*6.2. Data Applications*

Data management poses a considerable challenge for UAS-based stormwater inspections. The data obtained from UAS operations require substantial storage space, often overwhelming existing DOT platforms. To address this, DOTs should explore options to acquire cloud space from software vendors, enabling more efficient data storage. Furthermore, processing the acquired data can be complex, necessitating the development of automated data-processing methods. Automation can streamline data processing and enhance the usability of the acquired information. Additionally, establishing mechanisms for data sharing among relevant parties is crucial. Facilitating data sharing between field and office personnel ensures timely implementation of observations and identification of deficiencies by UAS inspectors. Revamping DOT computing capabilities or creating new services may not be feasible for all DOTs. Hence, considering commercial software solutions that offer cloud storage and data sharing among multiple users becomes a viable option. However, challenges concerning usability, safety, and access must be addressed.

Successfully overcoming these obstacles demands a combination of technical expertise, sufficient resources, standardized workflows, and adherence to legal and ethical guidelines. Research can also play a vital role in guiding DOTs on the best utilization of inspection data. Presently, acquired imagery and data are manually processed to identify stormwater BMP- and SCM-related deficiencies and other requirements. Implementing automation processes, such as AI, holds great potential for significantly improving data processing and identifying various needs efficiently.

*6.3. Regulation Limitations*

Existing Federal Aviation Administration (FAA) UAS regulations pose various challenges to the extensive utilization of UASs for stormwater inspections. These regulations restrict UAS operations over moving vehicles, mandate operators to maintain visual contact with the drone, and may limit flight options due to neighboring properties' restricted airspace. Consequently, operating UASs in urban environments becomes difficult. To overcome these limitations, DOTs may need to collaborate further with the FAA to obtain necessary waivers specifically for stormwater inspections. Additionally, the acceptance of UAS-acquired imagery and data for inspections by environmental agencies has been sluggish. To address this, DOTs should closely collaborate with state regulators and demonstrate the enhanced capabilities of UASs in stormwater inspection applications, surpassing traditional on-foot inspection methods. Building a strong case for the effectiveness and efficiency of UAS-based inspections can help gain greater acceptance from environmental agencies and pave the way for increased usage in stormwater management.

**7. Conclusions**

This study sought to identify and document DOT approaches for the use of UASs in conducting inspections of stormwater practices. Through a nationally distributed DOT survey questionnaire, that achieved an 80% response rate, and four case example interviews, this study investigates how UAS stormwater inspections have been deployed by DOTs and the strategies and programs that have been adopted or created.

The information gathered from this research sheds light on the following key aspects: (1) the use of UASs by DOTs for performing inspections of stormwater practices, (2) how DOTs incorporate UAS technologies into their stormwater inspection procedures, (3) staffing and equipment needs for successful UAS implementation, and (4) strategies for effectively managing and utilizing the datasets generated during UAS stormwater inspections. Overall, the study results provide insights into the current use of UAS technologies for stormwater BMP inspections and highlight potential areas for further development

and optimization. Findings from this research can be used to inform other domestic and international agencies on the current use of UASs for stormwater-related inspections in the U.S. transportation environment.

**Author Contributions:** Conceptualization, J.W., M.P. and R.S.; methodology, J.W., M.P. and R.S.; formal analysis, J.W., M.P. and R.S.; investigation, J.W., M.P. and R.S.; data curation, J.W.; writing—original draft preparation, J.W., M.P. and R.S.; writing—review and editing, J.W. and M.P.; project administration, M.P.; funding acquisition, J.W., M.P. and R.S. All authors have read and agreed to the published version of the manuscript.

**Funding:** This research was funded by the National Cooperative Highway Research Program (NCHRP) Synthesis, Project Number 20-05/Topic 53-09.

**Data Availability Statement:** Some or all data, models, or codes that support the findings of this study are available from the corresponding author upon reasonable request. The following datasets are available: spreadsheet database and results.

**Conflicts of Interest:** The authors declare no conflict of interest.

## Abbreviations

| | |
|---|---|
| AASHTO | American Association of State Highway and Transportation Offices |
| ADEM | Alabama Department of Environmental Management |
| AI | Artificial Intelligence |
| ASCE | American Society of Civil Engineers |
| BMP | Best Management Practice |
| CGP | Construction General Permit |
| DOT | Department of Transportation |
| FAA | Federal Aviation Administration |
| FHWA | Federal Highway Administration |
| GI | Green Infrastructure |
| GPS | Global Positioning System |
| LID | Low-Impact Development |
| LiDAR | Light Detection and Ranging |
| MS4 | Municipal Separate Stormwater Sewer System |
| NCHRP | National Cooperative Highway Research Program |
| PDF | Portable Document Format |
| RGB | Red Green Blue |
| SCM | Stormwater Control Measure |
| TRID | Transportation Research Information Documentation |
| TRR | Transportation Research Record |
| TS4 | Transportation Separate Storm Sewer System |
| UAS | Unmanned Aerial System |
| USEPA | United States Environmental Protection Agency |

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
