# Peer review of "Exploring the Integration of Unmanned Aerial System Technologies into Stormwater Control Inspection Programs"

_water, doi:10.3390/w15223924_

Round 1
Reviewer 1 Report
Comments and Suggestions for Authors
Please refer to the pdf file for comments

Author Response
The authors greatly appreciate the comments provided by the reviewer of this manuscript. This review has provided valuable insight into improving the overall quality of the paper. The authors agree that moving forward, UAS technologies can be of significant value with regards to stormwater inspection. Through future research we hope to develop operational guidelines that can be adopted by DOTs, and others, so that UAS technologies can be easily incorporated into stormwater inspection programs. All comments have been addressed and responses to individual comments are provided below. Track Changes was used to highlight edits made within the manuscript.
Respectfully,
Blake Whitman

Reviewer 2 Report
Comments and Suggestions for Authors
Title: Exploring the Integration of Unmanned Aerial System Technologies into Stormwater Control Inspection Programs.
Reviewer Comments:
1. Affiliation, Country, and PIN: Please ensure that the affiliation, country, and PIN details are provided.
2. Abstract: The abstract is well-structured, providing clear and concise information.
3. Keywords: Include more keywords without using abbreviations.
4. Please update the reference and request to add more references.
5. Introduction: Carefully revise the introduction section in accordance with the journal's guidelines or manuscript structure as outlined by Templet. The objective should be placed after the discussion of the background study. Utilize paragraphs as needed, but may omit section names like "Objective," "Background," "Construction Stormwater Management," or "Post-Construction Stormwater Management." Summarize the aforementioned sections in the introduction and concentrate on your study's focus and novelty in the final paragraph.
6. Methodology: Provide details about Recipient Identification and Survey Distribution addresses in a table, including a remarks column indicating whether a response was received or not. Additionally, include data storage analysis in a supporting information file.
7. Results and Discussion: Incorporate a separate discussion section to facilitate reader comprehension.
8. List of Abbreviations: Include a list of abbreviations before the reference list.
9. Conclusion: Revise the conclusion section for clarity and coherence.
Author Response
The authors greatly appreciate the comments provided by the reviewer of this manuscript. This review has provided valuable insight into improving the overall quality of the paper. All comments have been addressed and responses to individual comments are provided below. Track Changes was used to highlight edits made within the manuscript.
Respectfully,
Blake Whitman

Round 2
Reviewer 2 Report
Comments and Suggestions for Authors
Dear Authors,
Thank you very much for revising the manuscript in accordance with the reviewer's comments. I am now pleased to recommend your excellent work for publication in this journal.
Best wishes!!